# Diels–Alder Cycloaddition Reactions in Sustainable Media

**DOI:** 10.3390/molecules27041304

**Published:** 2022-02-15

**Authors:** Maria I. L. Soares, Ana L. Cardoso, Teresa M. V. D. Pinho e Melo

**Affiliations:** University of Coimbra, Coimbra Chemistry Centre–Institute of Molecular Sciences and Department of Chemistry, 3004-535 Coimbra, Portugal; ana.lucia.lopes@uc.pt

**Keywords:** Diels–Alder reaction, green solvent, bio-based solvent, deep eutectic solvents, water

## Abstract

Diels–Alder cycloaddition reaction is one of the most powerful strategies for the construction of six-membered carbocyclic and heterocyclic systems, in most cases with high regio- and stereoselectivity. In this review, an insight into the most relevant advances on sustainable Diels–Alder reactions since 2010 is provided. Various environmentally benign solvent systems are discussed, namely bio-based derived solvents (such as glycerol and gluconic acid), polyethylene glycol, deep eutectic solvents, supercritical carbon dioxide, water and water-based aqueous systems. Issues such as method’s scope, efficiency, selectivity and reaction mechanism, as well as sustainability, advantages and limitations of these reaction media, are addressed.

## 1. Introduction

The Diels–Alder reaction, a concerted [4 + 2] cycloaddition of a conjugated diene and a dienophile, is a powerful tool for the regio- and stereoselective construction of six-membered rings [1,2,3,4,5,6]. In its original version, two new C–C bonds are created in the Diels–Alder reaction and six-membered carbocyclic systems are obtained. Beyond this classical Diels–Alder reaction, other well-known types include the hetero-Diels–Alder reaction (at least one heteroatom is present on the diene or dienophile) leading to six-membered heterocycles and the intramolecular Diels–Alder reaction in which fused cyclic systems are obtained. Additionally, the Lewis acid catalyzed and organocatalytic Diels–Alder reactions have been developed, widening the scope of these reactions, namely the synthesis of optically active compounds via catalyzed asymmetric Diels–Alder reactions. Notably, a century after its discovery, the Diels–Alder reaction remains one of the most important green synthetic methodologies due to its theoretical 100% atom economy [7].

Driven by the green chemistry principles [8], the demand for sustainable and safe chemical processes over the last two decades has promoted the replacement of volatile organic compounds (VOCs) traditionally used as solvent media in synthetic chemistry with alternative green solvents [9,10,11,12,13,14] or even solvent-free conditions [15]. This endeavor gave rise to the introduction of new solvent media for chemical transformations featuring reduced environmental risk, reduced toxicity, reduced flammability, cost-effectiveness and reusability properties as the major advantages. Leading VOC alternatives for organic transformations are bio-based derived solvents (e.g., glycerol, lactic acid, gluconic acid) [16,17,18,19], liquid polymers (e.g., PEG) [20], ionic liquids [21,22], deep eutectic solvents [23,24,25,26], supercritical fluids [27,28,29,30,31], water and water-based aqueous systems [32,33,34,35,36,37]. The Diels–Alder reaction was no exception to this trend, leading to the advent of several studies, some of which were elegantly highlighted earlier by Wach and Brummond [38].

In this review, the most recent advances in the Diels–Alder reaction in environmentally benign and sustainable solvent systems since 2010 are covered, and [4 + 2] cycloaddition reactions performed in biomass-derived solvents, polyethylene glycol, organic carbonates, deep eutectic solvents, supercritical CO_2_ and water as solvent media are presented. Diels–Alder reactions in ionic liquids are not covered since reviews on this topic have recently been released [39,40].

## 2. Bio-Based Solvents

### 2.1. Glycerol

Glycerol, the main by-product in the biodiesel industry, is a nontoxic, biodegradable, recyclable and inexpensive viscous liquid. These properties, allied with the high stability, biocompatibility and ability to dissolve organic compounds poorly miscible in water as well as inorganic compounds (e.g., salts, acids, bases and transition metal complexes), make glycerol a valuable green solvent in synthetic organic chemistry [41,42,43].

The three-component aza-Diels–Alder reaction of substituted anilines, aldehydes and electron-rich alkenes, also known as three-component imino-Diels–Alder reaction, or multicomponent Povarov reaction (MCPR), is one of the most straightforward, efficient and atom-economical strategies towards complex cores starting from simple, inexpensive and available materials. This environmentally friendly methodology gives access to tetrahydroquinolines, quinolines and julolidines, which are scaffolds of great interest in synthetic organic and medicinal chemistry [44,45]. Perin and coworkers explored the intramolecular version of this reaction for the catalyst-free synthesis of octahydroacridines starting from (*R*)-citronellal (**1**) and substituted arylamines **2** using glycerol as a recyclable and eco-friendly solvent (Figure 1) [46]. Cycloadducts **3** and **4** were obtained as diastereoisomeric mixtures in good to high yields (75–98%) and moderate *cis*-selectivity when the reaction was carried out at 90 °C. Cycloadducts **3**/**4** (R = H) were obtained in lower yield (62%) using water as solvent, whereas the reaction carried out in organic solvents (DMSO, acetonitrile or ethanol) afforded the corresponding adducts in only trace amounts. Due to the insolubility of **3** and **4** in glycerol, products could be removed from the reaction medium by decantation, and the solvent could be reused for further aza-Diels–Alder reactions without loss of activity.

Glycerol was also used as an efficient reaction medium in the one-pot three-component aza-Diels–Alder reaction of anilines, aromatic aldehydes and cyclic enol ethers (Figure 2) [47]. 4-Aryl-furo[3,2-*c*]quinolines **7** were obtained selectively as *endo*-isomers from the reaction of 2,3-dihydrofuran (**6**), anilines **2** and aldehydes **5** in glycerol at 90 °C. The use of 3,4-dihydro-2*H*-pyran (**8**) as dienophile led to the formation of 5-aryl-2*H*-pyrano[3,2-*c*]quinolines as a mixture of *endo*- and *exo*-isomers **9** and **10**, respectively, and the *endo*-isomer was the major adduct. However, cycloadducts were extracted from the reaction medium using an organic solvent.

The domino Knoevenagel/hetero-Diels–Alder reaction (DKHDA) is a powerful synthetic tool for the construction of polycyclic heterocycles [48]. Parmar and coworkers reported the catalyst-free one-pot synthesis of heteropolycycles incorporating a thiochromeno[2,3-*b*]quinoline unit via a domino Knoevenagel/oxa-Diels–Alder reaction in glycerol [49]. The reaction of 2*H*-thiopyrano[2,3-*b*]quinoline-3-carbaldehydes incorporating an internal dienophile **11** and 5-pyrazolones **12**, carried out in glycerol at 120 °C over 3 h, furnished polyheterocycles **13**, with *cis*-fusion between pyran and six-membered carbocyclic rings, in high yields (82–93%) (Figure 3). The proposed mechanism involves the formation of the Knoevenagel alkene intermediate **15** mediated by glycerol via intermediate **14**, followed by intramolecular oxa-Diels–Alder reaction. Solvent studies reveal that the reaction is also feasible with organic solvents or under solvent-free conditions; however, under these conditions, the use of a catalyst is required. Moreover, it was demonstrated that glycerol can be recycled and reused five times.

The same group reported the one-pot synthesis of pyrano[3,4-*c*]chromene derivatives **18** from prenyl ether-tethered aldehydes **16** and acyclic or cyclic enolizable ketones **17** in glycerol through a DBU-catalyzed domino aldol/hetero-Diels–Alder domino sequence (Figure 4) [50]. Cycloadducts **18** with *cis*-fusion between the pyran rings were obtained in high yields (75–89%) when the reaction was carried out at 120 °C in the presence of DBU (25 mol%). The reaction involves an initial aldol condensation followed by an oxa-Diels–Alder reaction. Aldol intermediates with *E* geometry were isolated when using shorter reaction times, allowing the authors to conclude that the cycloaddition reaction occurs through the most favorable *endo*-*E*-syn transition state. It has been shown that glycerol can be reused at least three times without losing its activity.

### 2.2. Gluconic Acid

Gluconic acid (GA) can be obtained from biomass and possesses the ideal properties for being classified as a green and sustainable solvent (e.g., nontoxicity, biodegradability, recyclability, high boiling point, low vapor pressure) [51]. Due to the high solubility of gluconic acid in water, gluconic acid aqueous solutions (GAASs) have found wide application as solvent media for organic reactions, namely for the Knoevenagel condensation reaction. The Gu group reported the synthesis of 2*H*-pyrans by a one-pot multicomponent reaction between β-ketosulfones, formaldehyde and styrenes in a bio-based binary mixture solvent system composed of GAAS and a sugar-based organic base, meglumine [52]. The disclosed protocol involves the in situ generation of α-methylene-β-ketosulfones **21** through a Knoevenagel reaction of β-ketosulfones **19** and formaldehyde. Next, nucleophilic trap of **21** with styrenes **22** via oxa-Diels–Alder reaction afforded 2,6-diaryl-5-(phenylsulfonyl)-3,4-dihydro-2*H*-pyrans **23** in moderate to good yields (50–82%) (Figure 5). The binary solvent system GAAS/meglumine proved to play a pivotal role in controlling the selectivity of the hydroxymethylation step. Moreover, the hydrophilic properties of bio-based solvent meglumine allowed it to be easily recycled and reused in the GAAS/meglumine system without significant loss of activity.

## 3. Polyethylene Glycol

Polyethylene glycol (PEG), HO–(CH_2_CH_2_O)_n_–H, is a biodegradable, nontoxic, odorless, neutral, nonvolatile and inexpensive water-soluble polymer that has found widespread application as a green reaction medium for several organic transformations [20]. The Kouznetsov group reported the diastereoselective synthesis of heterolignan-like 6,7-methylendioxy-tetrahydroquinolines via a BF_3_^.^OEt-catalyzed three-component Povarov reaction using clove bud essential oil as a renewable raw material and PEG-400 as green solvent (Figure 6) [53]. Clove bud essential oil enriched with eugenol **24** (60.5%) was obtained by hydrodistillation of dried flower buds and then subjected to a solid base-catalyzed isomerization to give *trans*/*cis*-isoeugenol **25**, which could be used as a dienophile in the multicomponent hetero-Diels–Alder reaction without further purification. The reaction of **25** with aldimines generated in situ from substituted benzaldehydes **27** and 3,4-(methylendioxy)aniline (**26**) afforded *trans*-2,4-diaryl-1,2,3,4-tetrahydroquinolines **28** as racemic mixtures in moderate yields (35–55%). The reaction with phthalaldehydic acid (**29**) afforded isoindolo[2,1-*a*]quinolin-11(5*H*)-one **30** via an intramolecular condensation of the initially generated NH-tetrahydroquinoline core with the *o*-carboxylic acid function leading to the formation of the γ-lactam ring. It is noteworthy that these reactions also worked using acetonitrile as solvent media; however, less solvent volume and reduced reaction times were required when using PEG-400.

The intramolecular Povarov reaction of 2-aminoarylaldehydes bearing a tethered alkyne moiety with heterocyclic amines catalyzed by Amberlyst-15 in PEG-200 has been described [54]. This metal-free green protocol allowed the synthesis of [1,6]-naphthyridine-fused heterocycles **33** and **35** in good yields (60–81%) starting from 2-(*N*-propargylamino)-arylaldehydes **31** and 3-aminocoumarins **32** or 3-methyl-1-aryl-1*H*-pyrazol-5-amines **34**, respectively (Figure 7). The synthesis of these heterocyclic systems involves the initial formation of imine intermediate **36**, followed by an Amberlyst-15 promoted intramolecular [4 + 2] hetero Diels–Alder reaction as the key step. Deprotonation followed by air oxidation afforded the final products. It is worth mentioning the high atom economy of the protocol (over 96%) since water is the only by-product, as well as the use of an eco-friendly solvent and a recyclable catalyst.

Chandrasekhar and coworkers described the Diels–Alder reaction of cyclopentadiene (**39**) with substituted cinnamaldehydes **40**, catalyzed by (*S*)-TMS-diphenylprolinol (**41**) using PEG-400 as solvent (Figure 8) [55]. It was demonstrated that the *endo*/*exo* selectivity of the cycloaddition can be controlled by the addition of an acid cocatalyst which favors the formation of the *endo*-adduct. Thus, the reaction of cyclopentadiene (**39**) with cinnamaldehydes **40** in the presence of **41** as organocatalyst and perchloric acid as cocatalyst gave selectively *endo*-cycloadducts **42** as major products. The observed diastereoselectivity was rationalized by considering the involvement of the enamine complex intermediate **44** which was formed from cinnamaldehyde and the proline-derived organocatalyst and then enclosed in PEG-400 and stabilized by perchloric acid. The authors suggested that the cavity-like arrangement of the complex may be responsible for the facial selectivity which leads to the preferential formation of the *endo*-cycloadduct.

## 4. Organic Carbonates

Propylene carbonate (PC) is a polar aprotic solvent that can be obtained from propylene oxide and carbon dioxide, a renewable source of carbon, in a 100% atom economy reaction with relevance regarding the development of CO_2_ fixation processes. The noncorrosive, nontoxic, odorless and biodegradable properties of PC, allied with high boiling point, low vapor pressure and low cost, make this solvent a green and sustainable alternative to conventional organic solvents [56]. The Povarov reaction has also been explored using PC as an environmentally friendly solvent [57]. The one-pot iodine-catalyzed reaction of mono- or disubstituted anilines **2**, aromatic aldehydes **5** and isoeugenol (**45**) carried out at room temperature using PC as solvent medium afforded functionalized tetrahydroquinolines **46** in good to high yields (77–95%) and high diastereoselectivity (dr up to >99:1) (Figure 9). The same cycloadducts were obtained using organic solvents (e.g., dichloromethane, acetonitrile, toluene), albeit in low yields and requiring longer reaction times. It is noteworthy that, in general, products precipitated from the reaction medium and were purified by recrystallization.

Dimethyl carbonate (DMC) is a nontoxic methylating and/or methoxycarbonylating agent with wide application in the sustainable valorization of renewables [58,59]. Like PC, DMC can be synthesized by a green process using CO_2_ as a building block and can also be used as an eco-friendly solvent in several organic transformations. Ollevier and coworkers reported the Diels–Alder reaction of α,β-unsaturated carbonyl and *N*-acyloxazolidinone derivatives **47** with cyclopentadiene (**39**) using DMC as a green solvent (Figure 10) [60]. Among the solvents tested, from conventional organic solvents (dichloromethane and THF) to greener solvents (e.g., *N*-methylpyrrolidone, cyclopentyl methyl ether), DMC proved to be the most selective, affording cycloadducts with the highest yields and *endo*/*exo* ratio. Thus, the reaction of diene **39** and dienophiles **47**, catalyzed by the recyclable iron(ii) caffeine-derived ionic salt **48** and carried out in DMC, afforded cycloadducts **49** and **50** in good yields (up to 99%) and *endo*-selectivity. The reaction was also carried out in cyclohexa-1,3-diene; however, a longer reaction time was needed and the corresponding cycloadducts were obtained in low overall yield (20%) and 77:23 *endo*/*exo* ratio.

## 5. Deep Eutectic Solvents

First introduced by Abbot [61], deep eutectic solvents (DESs) are low melting mixtures obtained by combination of at least two components, a hydrogen bond acceptor (HBA), generally a quaternary ammonium or metal salt, and a hydrogen bond donor (HBD), to form a eutectic phase via hydrogen bond interactions. DESs are characterized by a melting point lower than those of the single components. The properties of DESs are very similar to those of room-temperature ionic liquids; however, the main difference from ionic liquids is that DESs also contain an organic molecular component, the HBD (e.g., urea, amide, polyol), generally as a major component. Due to their low vapor pressure, nonflammability, thermal and chemical stability, nontoxicity, biodegradability, recyclability and low price, DESs have emerged as green and sustainable media in different areas of chemical research [62,63,64,65], namely organic synthesis and catalysis [23,24,25,26].

In 2011, Nagare and Kumare reported a kinetic study of the Diels–Alder reaction between cyclopentadiene (**39**) and methyl acrylate (**51a**) using binary mixtures of urea, or methylated urea (DMU), with carbohydrates, or ternary mixtures of urea derivatives, NH_4_Cl and carbohydrates, demonstrating the potential of these eutectic mixtures as solvent media to accelerate this bimolecular reaction (Figure 11) [66]. They observed that the experimental rate constants were dependent on the percentage of urea in the carbohydrate melt; however, the best correlation was found to be with the solvent viscosity, which is in agreement with previous reports by the authors [67].

The Diels–Alder reaction of 9-anthracenemethanol (**54**) and *N*-ethylmaleimide (**55a**) in different solvent media under conventional heating and ultrasonic activation has been reported [68]. The influence of the solvent medium in the yield of the reaction was assessed by comparing DESs, obtained by the combination of different HBAs (e.g., choline chloride, methyltriphenylphosphonium chloride, (±)-menthol) and HBDs (e.g., glycerol (Gly), ethylene glycol (EG), urea), with water and organic solvents. In general, the Diels–Alder reaction of **54** and **55a** proved to be more efficient when carried out in eutectic mixtures than in conventional solvents (Figure 12). Moreover, it has been demonstrated that DESs could be recycled up to eight times without yield decrease and that the reaction time can be reduced from 24 h to 70 min when carrying out the reaction under ultrasound irradiation. The activating effect of DESs was attributed to the combined action of solvent viscosity, polarity and structure.

The effect of bio-based DESs on the *endo*/*exo* ratio of the Diels–Alder reaction of cyclopentadiene (**39**) and acrylates **51** has been recently described (Figure 13) [69]. DESs, prepared from racemic and optically enriched HBAs obtained from crude glycerol, proved to be more effective than organic solvents or ionic liquids, allowing cycloadducts to be obtained in higher yields and selectivity. The higher yields and *endo*-selectivity were achieved using lactic acid (LA) as the HBD of the eutectic mixture. However, a direct relationship between the chiral environment of the solvent and the observed selectivity was not demonstrated. It is noteworthy that cycloadducts were isolated from the reaction mixture by simple extraction with diethyl ether.

Garcia-Álvarez’s group reported a one-pot tandem cycloisomerization/Diels–Alder reaction using a ChCl (choline chloride)-based eutectic mixture as solvent [70]. The protocol involves the in situ generation of furans **59** by cycloisomerization of (*Z*)-enynols **57** using a ChCl/Gly (1:2) eutectic mixture as solvent and bis(iminophosphorane)-Au(i) complex **58** as catalyst (Figure 14). The Diels–Alder reaction of furans **59** with activated alkynes **60** afforded 7-oxanorbornadienes **61**, whereas the reaction with activated alkenes, **55b** or **62**, afforded selectively *exo*-7-oxanorbornenes **63**. The authors have demonstrated that complex **58** is crucial for the cycloisomerization step; however, it does not participate in the cycloaddition step.

The synthesis of 3,5-dimethylcyclohex-4-ene-1,2-dicarboxylates using a ChCl-based DES medium has been reported [71]. The methodology involves a Raney Ni-catalyzed hydrogenation of diacetone alcohol **64** followed by a one-step dehydration/Diels–Alder reaction of the resulting 2-methylpentane-2,4-diol **65** and diethyl fumarate (**67**) in a ChCl/EG deep eutectic mixture, catalyzed by Amberlyst-15, leading to the formation of compound **68** as a single product (Figure 15). Mechanistic studies demonstrated that the ChCl/EG and Amberlyst-15 play a role in the dehydration of **65**, affording 1,3-dienes **66a** and **66b** as a mixture (1:1.5). The observed selectivity was rationalized considering that the Diels–Alder reaction between **67** and **66b** is faster than that between **67** and **66a**, shifting the equilibrium from **66a** to **66b** and favoring the exclusive formation of **68**. Compound **68** and analogous derivatives could be further hydrogenated to give compounds that can serve as safe plasticizers for PVC materials.

Owing to their high reactivity and versatility, nitrosoalkenes and azoalkenes have been extensively used as building blocks for the synthesis and functionalization of an array of heterocycles. These electron-deficient heterodienes participate in inverse electron demand hetero-Diels–Alder reactions with a wide range of electron-rich heterocycles and nucleophilic olefins [72].

Pinho e Melo and coworkers disclosed the first example of an inverse electron demand hetero-Diels–Alder reaction in natural DESs [73]. Oxime- and hydrazone-bis(indolyl)methanes (BIMs) **71** were obtained as single *Z*-isomers via one-pot bis-hetero-Diels–Alder reaction of nitrosoalkenes and azoalkenes with indoles using a ternary mixture of water with ChCl/Gly (Figure 16). The protocol allowed the synthesis of BIMs in a more efficient and sustainable way than the previously reported strategies using DCM or a water/DCM solvent system [74,75]. The proposed mechanism involves the in situ generation of nitrosoalkenes or azoalkenes **73** by base-mediated dehydrohalogenation of α,α-dihalooximes or α,α-dihalohydrazones **69**, respectively, followed by hetero-Diels–Alder reaction with indoles to give cycloadducts **74**. Rearomatization of the indole unit triggers ring-opening of the six-membered ring affording 3-alkylindoles **75** bearing an α-halogenated oxime or hydrazone at the side chain. Dehydrohalogenation of **75** generates the corresponding 3-alkylheterodienes which react with another indole molecule, at the ***s-cis*** conformation, via a second hetero-Diels–Alder reaction to give BIMs **71** as single *Z*-isomers. In some cases, the formation of *E*/*Z*-isomeric mixtures of **71** resulting from an alternative mechanistic pathway via conjugated addition reactions was observed. When longer reaction times were used, carbonyl-BIMs **72** were obtained via a one-pot tandem hetero-Diels–Alder or conjugate addition/hydrolysis sequence.

Ochoa-Puentes and coworkers explored the multicomponent Povarov reaction for the one-pot synthesis of quinoline derivatives using zinc chloride based eutectic mixtures as reaction media [76,77]. Tetrahydroquinolines **77** were obtained with *endo*-selectively by the reaction of *trans*-anethole (**76**), the major component of star anise oil obtained by hydrodistillation of anise seeds, with anilines **2** and aromatic aldehydes **5** using a ChCl/ZnCl_2_ eutectic mixture (Figure 17a) [76]. On the other hand, the reaction of anilines **2**, aromatic aldehydes **5** and indene (**78**) performed in a urea/ZnCl_2_ eutectic mixture led to the synthesis of indeno[2,1-*c*]quinolines **79** in high yields (89–99%) (Figure 17b) [77]. The proposed mechanism for the formation of quinolines involves the initial reaction of amine **2** with DES-activated aldehyde to give *N*-aryl imine (Schiff’s base) **80** followed by DES activation to generate intermediate **81**. The *endo*-selectivity could be rationalized considering that **81** participates in a [4 + 2] concerted cycloaddition via a *trans*-*endo*-favored transition state followed by a 1,3-H shift. However, the most plausible mechanism was rationalized considering a two-step ionic mechanism involving an *endo*-selective nucleophilic dienophile attack to imine **81** followed by an intramolecular Friedel–Crafts reaction to give selectively the 2,4-*cis*-isomer **77**. In the formation of indenoquinolines **79** an additional aerobic oxidative dehydrogenation was proposed. It is noteworthy that for both synthetic procedures, DESs were reused up to three cycles and green metric analysis demonstrated that the synthetic strategy has a favorable environmental impact.

DESs have also been successfully employed as environmentally friendly reaction media for the covalent functionalization of carbon nanomaterials via catalyst-free Diels–Alder click reaction. In 2018, Lim’s group reported the functionalization of multiwalled carbon nanotubes (MWNTs) with polymeric materials containing ionic liquids (ILs) in their repeating units [78]. The Diels–Alder reaction of MWNTs **83** and imidazolium-based poly(ionic liquid)s featuring furfuryl moieties (P(F-ILs)) **84** was performed in a Ch/EG eutectic mixture under ultrasound irradiation at 60 °C (Figure 18). Functionalized MWNTs **85** were obtained with shorter reaction times and higher grafting density of P(F-ILs) on MWNTs when compared with the reaction in water. Recently, the same strategy was applied for the covalent functionalization of reduced graphene oxide (rGO) with P(F-ILs), delivering a new rGO-P(F-IL) hybrid material with good electrochemical capacitor behavior [79].

## 6. Supercritical Carbon Dioxide

Supercritical carbon dioxide (scCO_2_) is an abundant, low-cost, nontoxic and nonflammable fluid. The physical properties of scCO_2_ are intermediate between the gas and the liquid phases. These properties can be tuned by changing pressure and temperature; in particular, changes close to the critical point enable drastic changes in density, viscosity and diffusion. Among supercritical fluids (SCFs), scCO_2_ has received special attention since it is readily accessible at a low critical temperature (T_c_ = 31 °C) and moderate critical pressure (P_c_ = 75.8 bar) [80,81]. In addition, scCO_2_ has the ability to dissolve organic compounds and can be easily removed from the reaction mixture, making it a sustainable alternative to conventional organic solvents in synthetic transformations [27,28,29,30,31]. Keshtov et al. disclosed a green approach for the synthesis of photoluminescent polymers based on phenyl-substituted polyfluorenes using scCO_2_ as the solvent medium [82]. Phenylated polyfluorenes **88** were synthesized through a catalyst-free Diels–Alder reaction of fluorene-containing bis(tetraarylcyclopentadienone) monomer **86**, acting as diene, with bis(acetylenes) **87** acting as 2π-component (Figure 19). Phenyl-substituted polyfluorenes synthesized using scCO_2_ as solvent showed similar properties to those synthesized using chloronaphthalene as solvent, demonstrating that scCO_2_ is a suitable alternative to organic solvents for the synthesis of phenylated polyfluorenes via the Diels–Alder reaction.

The Diels–Alder reaction of cyclopentadiene (**39**) and 1,3-butadiene (**89**) in scCO_2_ has been reported, allowing the synthesis of 5-vinyl-2-norbornene **90**, a precursor of 5-ethylidene-2-norbornene which is a valuable monomer used in the manufacture of ethylene propylene rubber (Figure 20) [83]. The major drawback of this reaction is the reactivity pattern of the starting materials, cyclopentadiene and 1,3-butadiene, that can act as both dienes and dienophiles in the Diels–Alder reaction and therefore give rise to a panoply of by-products (e.g., **91**). The Diels–Alder reaction of **39** and **89** was carried out in neat conditions, in the presence of organic solvents and in scCO_2_. The most satisfactory yields and selectivities were achieved when using scCO_2_ as the solvent medium, providing the target compound **90** in 25% yield, by-product **91** in 17% yield and other unidentified compounds in 6% yield. The observed slight increase in selectivity was rationalized considering the unique physical properties of scCO_2_ acting as a thinner and allowing the suppression of eventual side reactions, being an alternative to the commonly used large amounts of solvents and/or inhibitors.

Ito and coworkers disclosed a protocol for the synthesis of isoindoles from bicyclopyrroles by a retro-Diels–Alder reaction in scCO_2_ [84]. In general, bicyclopyrroles were converted into the corresponding isoindoles in high yields; however, in some cases, the instability associated with the exomethylenic structure causes the decomposition of the isoindole derivatives. In these cases, the formation of the retro-Diels–Alder product in the reaction medium was confirmed by performing the retro-Diels–Alder reaction in the presence of a trapping agent for isoindole. Thus, the reaction of *N*-tosylated bicyclopyrrole **92** in the presence of *N*-phenylmaleimide (**55c**) carried out in scCO_2_ (20 MPa) at 185 °C over 210 min led to the formation of the *exo*-Diels–Alder adduct **94** in 72% yield, which confirms the generation of isoindole **93** under the reaction conditions (Figure 21).

## 7. Water

The use of water as an alternative benign and sustainable solvent in promoting Diels–Alder cycloaddition reactions has been widely explored since the seminal work of Breslow’s and Grieco’s groups [85,86,87,88]. The remarkable rate enhancements and selectivities achieved on moving from conventional organic solvents to aqueous conditions are the major driving force behind the growing interest in the study of the Diels–Alder reaction in this medium [37,38].

### 7.1. The Water Effect: Experimental and Theoretical Studies

Hydrophobic interactions, along with hydrogen bonding, polarity and cohesive energy density, are the major factors responsible for the rate enhancement in “on-water” Diels–Alder cycloadditions. Several systematic experimental and theoretical studies analyzing the pivotal role of water in the rate enhancement and selectivity of the Diels–Alder cycloaddition reactions have been disclosed. Shrinidhi reported a comparative study of the rates and efficiency of the Diels–Alder cycloaddition reaction between cyclopentadiene (**39**) and electron-poor alkenes **95** in solvent media featuring different polarities (organic solvents and water) (Figure 22) [89]. For each dienophile, it was observed that cycloadducts **96** yield increases with the increase in solvent polarity (hexane < dichloromethane < THF–water < water) and concomitantly with the solvophilicity of the dienophile. The effect of adding a catalytic amount of water along with organic solvents (dichloromethane and THF) was also explored, demonstrating that the reaction is more efficient in the presence of catalytic amounts of water than in pure organic solvents. Based on these results, combined hydrophobic and hydrogen bonding effects were attributed to the rate enhancement of the Diels–Alder reaction when carried out in water as compared to conventional organic solvents. The hydrogen bonding effect was previously reported by Domingo’s group supported by a density functional theory (DFT) study on the catalytic effect of water in the intramolecular Diels–Alder reaction of a quinone system [90]. Theoretical calculations demonstrated the pivotal role of water in stabilizing the polarized transition state through hydrogen bond formation with the internal electron-deficient dienophile.

The Vilches-Herrera group reported an experimental and theoretical study on the influence of noncovalent interactions in the stereo- and regioselectivity of the aza-Diels–Alder reaction between 5-aminopyrrole-derived aldimines and electron-poor dienophiles [91]. Depending on the nature of substituents present on both diene and dienophile, the reaction of aldimines **97** and electron-poor styryl substrates **98**, carried out in water under MW irradiation over 20 min, afforded regio- and stereoselectively the *exo*-adducts of tetrahydro-1*H*-pyrrolo[2,3-*b*]pyridines **99**, or 7-azaindoles **100**, also in a regioselective fashion (Figure 23). The reaction is also compatible with alkynes (dimethyl acetylenedicarboxylate) and monosubstituted alkenes (ethyl acrylate and acrylonitrile); however, in the latter case, mixtures of regioisomers were obtained. The experimental results were corroborated by DFT calculations which suggest that the regioselectivity and the unusual *exo*-selectivity were controlled by noncovalent attractive π–π interactions at the TS region instead of secondary orbital interactions. In the absence of noncovalent interactions (e.g., with ethyl acrylate), thermodynamic control prevails and regioisomeric mixtures were obtained.

A comparative study on the rates of the Diels–Alder reaction of 9-hydroxymethylanthracene (**54**) and *N*-phenylmaleimide (**55c**) in water, ethylene glycol and conventional organic solvents (butan-1-ol, acetonitrile, chloroform and 1,4-dioxane) has been reported (Figure 24) [92]. Kinetic measurements, performed at 45 °C using spectrophotometric methods, showed that, under homogeneous conditions, no relevant increase in the reaction rate on going to protic solvents (butan-1-ol) or to more polar ones (e.g., acetonitrile) was observed. On the other hand, for the heterogeneous Diels–Alder reaction, carried out under vigorous stirring, a significant acceleration was observed, attributed to the reactants’ activation on the water–organic phase boundary.

The kinetics of the Diels–Alder cycloaddition reactions in water can be modified by the presence of inorganic salts, which can change the hydrophobic interactions involving the diene and dienophile, the internal pressure (by changing the cohesive energy) and the solvation of the transition states. The Kumar group studied the influence of quaternary ammonium salts in the cycloaddition of 9-hydroxymethylanthracene (**54**) and *N*-ethylmaleimide (**55a**) in water, concluding that higher concentrations of salt in the reaction media lead to a deceleration of the reaction rate [93]. This was rationalized by the reduction in hydrophobic interactions, responsible for bringing the reactants closer, which were progressively blocked upon the continuous increase in the concentration of salt. They also concluded that ammonium salts bearing long alkyl groups promote the dissolution of the nonpolar reactants by rearranging water molecules up to a second solvation shell around the dissolved salts, thus behaving as anti-hydrophobic or salting-in agents. In another study, Shimizu and coworkers investigated the effect of several rate-enhancing salts (e.g., LiCl, guanidinium sulfate) and rate-reducing salts (e.g., NaClO_4_, guanidinium acetate) for the same reaction, demonstrating that the interactions between ions and diene **54** are crucial for controlling the reaction rate [94]. A rate reduction was observed for the reaction in the presence of rate-reducing salts, which show a preferential interaction with diene, and is consistent with the increased **54** solubility. On the other hand, the major driving force behind the rate enhancement with rate-enhancing salts is the salting-out of the hydrophobic diene **54** by the rate-enhancing salt, which exhibits a stronger affinity with the transition state. The effect of LiCl on the Diels–Alder reaction of cyclopentadiene (**39**) and methyl vinyl ketone in water was also investigated, demonstrating a rate enhancement in the presence of this salt arising from the destabilization of the reactants by hydrophobic effect resulting in the decrease in the activation barrier [95]. The instability associated with the hydrophobic interaction in the transition state complexes is suppressed by the formation of hydrogen bonds, which are stronger compared to those of the dienophile.

Kumar and coworkers reported a Diels–Alder cycloaddition reaction of cyclopentadiene (**39**) with methyl acrylate (**51a**) using a recyclable and highly viscous supersaturated water-based solvent obtained by supersaturation of water (~18% *w*/*w*) with carbohydrates, an organic acid and an organic ketone [96]. The second-order rate constant (*k*_2_, 7.92 × 10^−5^ M^−1^ s^−1^) determined for the reaction carried at room temperature is about 26-fold higher than that obtained for the same reaction in water. This rate enhancement was attributed to the combined presence of OH groups from the carbohydrates, which allows multiple hydrogen bonding sites, and the hydrophobic effect by water molecules. It is noteworthy that it is also possible to carry out the reaction of cyclopentadiene (**39**) and methyl acrylate (**51a**) using eutectic mixtures of carbohydrates as solvent medium; however, to ensure these mixtures were in a liquid state, higher temperatures (>85 °C) were required (cf. Figure 11) [66,96].

### 7.2. Noncatalyzed Diels–Alder Cycloaddition Reactions

Romám and coworkers reported a catalyst-free protocol for the Diels–Alder cycloaddition of water-insoluble furan derivatives and *N*-substituted maleimides using “on-water” conditions [97]. The reaction of furans **102** and maleimides **55** carried out at room temperature afforded cycloadducts **103** and **104** in quantitative yields and *endo*-selectivity (Figure 25a). When the reaction was carried out at 65 °C, the reaction times were shortened and an increase in the *exo*-adducts was observed. On the other hand, the reaction between furfural *N*,*N*-dimethylhydrazone (**105**) and maleimides **55** afforded selectively the *exo*-adducts **106** obtained as single products, mixtures of **106** and phthalimides **107**, or **107** as the sole products, depending on the maleimide precursors and reaction conditions (Figure 25b). Upon heating, cycloadducts **106** can be converted into the corresponding phthalimides **107**, whose formation was rationalized considering the ring-opening of the *exo*-7-oxabicycle followed by dehydration. A comparison with previously reported conditions for the same reaction, using organic solvents or solvent-free conditions, demonstrated the advantages of the “on-water” methodology, namely shorter reaction times, higher yields, simple workup, milder reaction conditions and the absence of a catalyst.

The aqueous Diels–Alder cycloaddition reaction of bio-based furans derived from renewable feedstocks represents a valuable example of a green and sustainable process in organic synthesis. Hailes and coworkers reported the synthesis of phthalimide-hydrazones **110** via an uncatalyzed one-pot tandem reaction in water starting from substituted furfurals **108** (Figure 26a) [98]. The three-step cascade protocol involves the initial in situ formation of hydrazones **109** by reacting **108** and *N*,*N*-dimethylhydrazine, cycloaddition reaction with maleimides **55** and subsequent aromatization. The cycloaddition/aromatization also works for the reaction of hydrazone **109a** with non-maleimide dienophiles **111**, acrylonitrile and fumaronitrile, affording benzonitrile-hydrazones **112** in moderate yields (24–68%) (Figure 26b). The one-pot three-step reaction of furfural **108a** (R^1^ = R^2^ = R^3^ = H) in the presence of dimethyl maleate afforded the corresponding cycloadduct, albeit in low yield.

Ananikov and coworkers disclosed a three-step synthesis of polycyclic compound **116** from bio-based 5-(hydroxymethyl)furfural (HMF) (**113**) involving a Diels–Alder cycloaddition step (Figure 27) [99]. Computational calculations demonstrated that the Diels–Alder reaction of 2,5-bis(hydroxymethyl)furan (BHMF) (**114**) and maleimide is energetically more favorable than that of HMF (**113**) with the same dienophile. Thus, HMF (**113**) was reduced to BHMF (**114**) in aqueous solution. Subsequent cycloaddition reaction of **114** with maleimide (**55d**) in water furnished compound **115** diastereoselectively (96% *de*) as the *endo*-adduct in 67% yield. To prevent the retro-Diels–Alder transformation, **115** was further reduced, giving compound **116** in 87% yield (58% overall yield). It is noteworthy that the three-step one-pot protocol in water, starting from HMF (**113**), proved to be more efficient, allowing the synthesis of **116** in 73% yield.

Recently, the Bruijnincx group disclosed the Diels–Alder reaction of furfurals with water-soluble maleimides in aqueous medium [100]. The reaction of furfurals **117** with maleimides **55** at 60 °C in water furnished cycloadducts **118** and **119** in low to moderate yields (1–58%) with *exo*-selectivity (Figure 28a). Different selectivity was observed for the Diels–Alder reaction of HMF (**113**) with a maleimide, which led to the formation of the *endo*-cycloadduct as the major product. It is noteworthy that for more lipophilic adducts (R^2^ ≠ H), an increased preference for dehydration of the geminal diol **118**/**119** back to the aldehydes **120**/**121** and partitioning to the organic phase was observed. DFT calculations demonstrated that the limitation of the Diels–Alder reaction of furfural, and other related electron-poor furans, is of thermodynamic rather than kinetic nature and can be circumvented by using water, which acts both as solvent and as reactant in the geminal diol formation. Hydration of the formyl group is thermodynamically possible whether it occurs prior to or after the cyclization step, by favoring the rate of the direct reaction or decreasing the rate of the retro-Diels–Alder reaction, respectively.

The same group reported the use of oxidized furfurals, furoic acids, as dienes in the aqueous Diels–Alder cycloaddition reaction with maleimides [101]. The base-mediated reaction of furoic acids **122** and maleimides **55**, carried out in water under mild conditions, led to the formation of cycloadducts **123**/**124** in moderate to good yields and *exo*-selectivity (Figure 28b). In this case, the cycloaddition reaction benefits from the rate enhancement caused by the water effect as well as from the activation of the 2-furoic acids by conversion to the corresponding carboxylate salts. Notably, the reaction also works well when using 2-furoic acid esters or furamides as dienes, affording selectively *exo*-adducts as major products in moderate to good yields. It is noteworthy that when 2-furoic acid esters are used as dienes, the system is no longer homogeneous and reactions proceed “on-water”.

Ohmic heating (ΩH)-assisted synthesis has emerged as a new methodology in organic synthesis, circumventing some of the limitations presented by the microwave irradiation (MW) and conventional heating methods. The advantages of ΩH-assisted synthesis were illustrated by the Diels–Alder cycloaddition of 9-hydroxymethylanthracene **54** with *N*-methylmaleimide (**55b**) under three different heating processes: conventional, MW and ohmic heating (Figure 29a) [102]. After only two minutes of reaction, cycloadduct **125** was obtained in 80% yield using ohmic heating, whereas under microwave irradiation the yield was significantly lower (53%). The conventional heating conditions also afforded the desired product in relatively lower yield (57%).

Ohmic heating was also successfully applied in the regioselective and *site*-selective synthesis of coumarinyl porphyrin derivatives (Figure 29b) [103]. The sequential Knoevenagel condensation of 4-hydroxycoumarin **126** and aromatic aldehydes **127** leading to α-methylenechromane derivatives **128**, followed by hetero-Diels–Alder reaction of these in situ generated dienes with 2-vinyl-5,10,15,20-tetraphenylporphyrinatozinc(ii) (**129**), was performed in aqueous medium using ohmic and conventional heating. The ΩH-assisted synthesis provided the coumarinyl porphyrin derivatives **130** in higher yield and shorter reaction time. The higher efficiency of these transformations under ohmic heating was rationalized as resulting from the high heating rates achieved in the beginning of the reactions which may lead to a more uniform heating and to less decomposition of the reactants. Additionally, the electrical dynamic perturbation in ohmic heating may also influence the polarization of the reaction medium and improve the transport properties.

A series of diversely substituted 6*H*-benzo[*c*]chromenes **134** have been prepared via intramolecular Diels–Alder reactions of furan with unactivated alkenes in an aqueous medium under MW irradiation (Figure 30) [104]. The reaction pathway involves the initial formation of cycloadduct **132** which undergoes ring-opening to give intermediate **133**, followed by aromatization to yield the desired 6*H*-benzo[*c*]chromenes **134** in yields ranging from 45 to 85%. This methodology was successfully extended to the synthesis of 6*H*-benzo[*c*]chromen-8-ols by employing 2-(2-(propargyloxy)phenyl)furan derivatives as reactants; however, the use of a mixture H_2_O/EtOH (4:1) as reaction medium was required to attain good efficiency.

The use of a thin film microfluidic platform, namely a vortex fluidic device (VFD), to promote Diels–Alder reactions between 9-substituted anthracenes **135** and *N*-substituted maleimides **55** in media with a high mole fraction of water, without the need for catalysts, has been disclosed by Raston and coworkers (Figure 31) [105]. Using the confined mode of VFD processing, a dynamic thin fluid film is formed on the walls of the tube where there is intensive shear, with high mass and heat transfer, thus accelerating the Diels–Alder reaction by providing a constant “soft energy”. Carrying out the reactions under the optimized VDF processing parameters (e.g., 5000 rpm, tilt angle *θ* = 45°), at 50 °C in 10% ethanol in water, led to the formation of the expected cycloadducts **136** in good to excellent yield, after 30 min of reaction. In contrast to previous studies, in which a higher speed favored the reaction progress, it was observed that changing the speed had little effect on the reaction outcome. In fact, for rotations higher than 5000 rpm, a slight decrease in the yield was observed.

A synthetic methodology towards dipyrromethanes involving two consecutive Diels–Alder reactions of azoalkenes and pyrrole has been accomplished by Pinho e Melo and coworkers (Figure 32) [106]. The base-mediated dehydrohalogenation of alkyl and aryl α,α-dihalohydrazones **69** originates azoalkenes **73** and **140** which undergo a hetero-Diels–Alder reaction with pyrrole to give selectively the desired dipyrromethanes **138** in moderate to good yields. The “on-water” reaction conditions afforded the target products in higher yields with significantly shorter reaction times and simpler purification procedures than carrying out the reaction in dichloromethane or in solvent-free conditions. This one-pot methodology was further applied to the synthesis of dipyrromethanes derived from α,α-dihalooximes and pyrrole.

The reactivity of phosphorylated nitrosoalkenes towards enol ethers and pyrrole has been explored by Palacios and coworkers (Figure 33) [107,108]. The one-pot synthetic methodologies gave access to functionalized 4-phosphorylated 1,2-oxazines **144** and open-chain 2-substituted pyrroles **145** through the treatment of the α-halooxime precursors **141** with base, followed by [4 + 2] cycloaddition reaction of the in situ generated **142** with enol ethers and pyrrole, respectively, in a regioselective fashion. The formation of pyrroles **145** was rationalized considering the rearomatization of pyrrole unit of the firstly formed cycloadducts. The substrate scope includes nitrosoalkenes bearing a variety of substituents at C-3 (Me, Et, Ph and CO_2_Me) and C-4 (P(O)Ph_2_, P(O)(OEt)_2_). Once again, the “on-water” protocol showed higher chemical efficiency and selectivity than the solvent-free conditions or the use of organic solvents, leading to 1,2-oxazines **144** with excellent diastereoselectivity.

A chemo- and diastereoselective synthetic multicomponent route to pyrazolo-tetrahydropyridines involving an intramolecular aza-Diels–Alder reaction was disclosed by Shaabani and coworkers (Figure 34) [109]. A plethora of pyrazolo-tetrahydropyridines **151** was efficiently prepared from benzoylacetonitrile derivatives **146**, hydrazines **147**, aromatic aldehydes **150** and styrenesulfonyl or cinnamoyl chloride in water in the presence of base, using the group-assisted purification (GAP) chemistry strategy. In this one-pot procedure, 5-amino-pyrazoles **148** were initially formed from benzoylacetonitrile derivatives **146** and hydrazines **147** through tandem condensation and thermal cyclization, under solvent-free conditions at 120 °C. Then, intermediates **148** reacted chemoselectively with 2-formylphenyl-(*E*)-2-phenylethenesulfonate derivatives (e.g., **154**), generated in situ from **149** and **150**, to give intermediate **155** which undergoes an intramolecular aza-Diels–Alder reaction via *exo*-approach to yield pyrazolo-tetrahydropyridines **151** as single diastereoisomers (Figure 34). The proposed reaction mechanism was corroborated by some control experiments in which the synthesis of an intermediate (e.g., **155**) was accomplished.

Vilches-Herrera and coworkers developed a microwave-assisted methodology for the preparation of annulated tetrahydropyridines by intramolecular aza-Diels–Alder reaction (Figure 35) [110]. Featuring several requirements for sustainable chemistry such as the use of water as solvent, microwave irradiation, absence of catalyst and easy product isolation by precipitation, it gives access to important scaffolds bearing the tetrahydropyridine and the chromane moieties. Under the optimized reaction conditions at 200 °C, a range of alkenyl 2-iminopyrroles and 2-iminopyrazoles **156** underwent intramolecular aza-Diels–Alder reaction giving rise to the corresponding dihydrochromeno-pyrrolo- and dihydrochromeno-pyrazolo-tetrahydropyridines **157** in moderate to high yield and with overall excellent *trans*-diastereoselectivity. Moreover, when propargylic derivatives were used as dienophiles, aromatic annulated pyridines **158** were isolated in good yields. As observed for other aza-Diels–Alder reactions, the stereoselectivity of the process was strongly dependent on the solvent used, with selective formation of *trans*-**157** when reactions were performed in water, whereas the use of nonpolar solvents such *p*-xylene led to a mixture of *cis*/*trans*-**157** products.

A multicomponent chemo-, regio- and stereoselective approach to spiro{pyrazolo[1.3]-dioxanopyridine}-4,6-diones and spiro{isoxazolo[1.3]-dioxanopyridine}-4,6-diones derivatives **161** carried out in water under microwave irradiation was reported by Tu and Li (Figure 36a) [111]. Using Meldrum’s acid (**159**), a wide variety of aromatic aldehydes and 3-methylisoxazol-5-amine (**160a**) or 3-methyl-1-phenylpyrazol-5-amine (**160b**) as substrates, the target products **161** were obtained in high yield within a very short reaction time (9–13 min). The key step in this domino reaction involves the hetero-Diels–Alder reaction between imine **162** and Knoevenagel adduct **163**, which follows the *endo* rule and gives rise to *syn*-**161** as single stereoisomers. Interestingly, a different chemoselectivity was observed when N-H- and N-Me-pyrazol-5-amines were employed, leading to pyrazolo[3,4-*b*]pyridines in high yields. This outcome resulted from a different reaction pathway in which a Michael addition of pyrazole’s amino group onto intermediate **166** occurs, followed by addition to the carbonyl favored by the higher reactivity of the amino group. This protocol was further applied to the synthesis of pyrimidinespiroisoxazolo[5,4-*b*]pyridines **165** (Figure 36b) [112]. Under the same reaction conditions, barbituric acids **164**, two equivalents of aromatic aldehydes and 3-methylisoxazol-5-amine (**160a**) provided the desired *syn*-**165** in yields ranging from 78 to 89%.

Majundar and coworkers explored the reactivity of *O*-propargylated salicylaldehydes **167** towards 1-methylindoline-2-thione (**168a**) and 4-hydroxydithiocoumarin **170**, under catalyst-free and aqueous medium conditions, uncovering synthetic routes to indole-annulated [6,6]-fused thiopyranobenzopyrans **169** and benzopyran-annulated thiopyrano[2,3-*b*] thiochromen-5(4*H*)-ones **171**, respectively (Figure 37) [113,114]. The mechanism involves initial Knoevenagel condensation to give the heterodiene intermediates **172** or **173**, followed by an intramolecular aza-Diels–Alder reaction to afford the corresponding heteropolycyclic compounds in high yield. The high efficiency of these catalyst-free transformations, as well as the observed regioselectivity in the synthesis of **171**, was attributed to the presence of the softer sulfur atom in the diene moiety. This reactivity was further extended to *O*-allylsalicylaldehydes which under the same reaction conditions originated *cis*-annulated [6,6]-fused thiopyrano benzopyran derivatives in a highly regio- and stereoselective fashion [115].

Moghaddam and coworkers demonstrated that *O*-acrylated salicylaldehydes **174** can also undergo DKHDA with dihydroindole-2-thiones (e.g., **168a**) in water under reflux to afford polycyclic indole-annulated thiopyranocoumarin derivatives, in good to high yields with high regio- and stereoselectivity [116]. However, a different outcome was observed in the DKHDA of these dienophiles **174** with 4-hydroxydithiocoumarin **170** (Figure 38) [117]. Although the efficiency of the domino reaction remained high, yielding thiochromone-annulated thiopyranocoumarin **175** and **176** in good overall yield, the diastereoselectivity of the domino reaction varied significantly with the substituents on the *O*-acrylated salicylaldehydes, namely the substituent on the carbon–carbon double bond (R^3^). Thus, methyl-substituted *O*-acrylated salicylaldehydes **174** afforded selectively *cis*-**176** via an *endo*-transition state, whereas phenyl-substituted *O*-acrylated salicylaldehydes **174** gave predominantly *trans*-**175** resulting from an *exo*-transition state.

A similar synthetic strategy has been described for the preparation of a wide variety of benzo-δ-sultones bearing hexahydro-chromene (e.g., **178**), tetrahydro-pyrano[2,3-*d*]pyrimidine (e.g., **179**) and thiopyrano indole (e.g., **180**) motifs (Figure 39) [118,119]. Initial Knoevenagel reaction of aldehydes **177** with dimedone **149**, *N*,*N*-dimethylbarbituric acid (**164a**) or indoline-2-thiones **168** generates the dienophile tethered to the corresponding diene moiety by a sulfonate link, which undergoes intramolecular hetero-Diels–Alder reaction to give the desired annulated benzo-δ-sultone derivatives in moderate to high yield, with good to high diastereoselectivity. As in the aforementioned methodologies, the reactions were carried out in refluxing water under catalyst-free conditions, affording the *cis*-*trans*-annulated cycloadducts as major isomers.

Recently, the domino Knoevenagel/hetero-Diels–Alder reaction of (*E*)-*N*-alkyl-2-aryl-*N*-(2-formylphenyl)ethane-1-sulfonamides **181** with indoline-2-thiones **168** in water has been disclosed by Langer and Kiamehr (Figure 40) [120]. After 5 h under refluxing conditions, the resulting pentacyclic benzosultam-annulated thiopyranoindole derivatives **182** and **183** were obtained regioselectively in high yields albeit with poor stereoselectivity. The *N*-methyl indoline-2-thione led to preferential formation of the *cis*-isomers **182**, whereas the *N*-ethyl indoline-2-thione afforded the *trans*-**183** as major products. This was suggested to result from the higher steric hindrance of the ethyl group which favored the *exo*-transition state.

A series of cyclic 1,3-dicarbonyls, namely dimedone, *N*,*N*-dimethylbarbituric acid, 1,3-indanedione, 4-hydroxy-6-methyl-2*H*-pyran-2-one and chroman-2,4-diones, underwent a catalyst-free domino Knoevenagel/hetero-Diels–Alder reaction with (*E*)-*N*-(2-formylphenyl)-*N*-methylcinnamamide derivatives in water, giving access to a variety of tetra- and pentacyclic dihydroquinolinones annulated with diverse cyclic motifs in high yields, albeit with low stereoselectivity in most cases (Figure 41) [121]. Nevertheless, the DKHDA reaction of **184** with chroman-2,4-diones **185** proceeded selectivity via *exo*-transition state yielding the *trans*-**186** as single diastereoisomers.

The one-pot three-component synthesis of a range of pentacyclic-fused pyranochromenone and pyranoquinolinone benzosultone derivatives was successfully accomplished by Ghandi and coworkers (Figure 42) [122]. The developed synthetic methodology involved successive addition of 2-hydroxybenzaldehydes **187**, 4-hydroxycoumarins **185** and a catalytic amount of EDDA to an aqueous solution of styrenesulfonyl chloride and K_2_CO_3_, followed by refluxing conditions. The reactions proceeded smoothly through *O*-sulfonylation/Knoevenagel condensation/hetero-Diels–Alder reaction cascade affording *cis*-*trans*-annulated coumarins **188** as major products in good to high yields. Similarly, when 4-hydroxy-2-quinolones were used, *cis*-*trans*-pyranoquinolinone benzosultone derivatives were obtained selectively, thus demonstrating that the hetero-Diels–Alder reaction proceeds via *endo*-*E*-*syn* transition states.

A range of pyrano[2,3-*d*]pyrimidines were efficiently prepared through a three-component one-pot domino Knoevenagel/Diels–Alder reaction in aqueous suspension (Figure 43) [123]. Reactions of barbituric acids, aldehydes and ethyl vinyl ether were performed at room temperature, while the use of a styrene derivative or *N*-vinyl-2-oxazolidinone as dienophiles required increasing the reaction temperature to 60 °C. These transformations were highly diastereoselective, furnishing preferentially or exclusively the *endo*-*cis*-annulated uracils **191** and **192** in high yields. Along with pyrano[2,3-*d*]pyrimidines **192**, 5-methyl-substituted derivatives **193** arising from the reaction with in situ generated acetaldehyde were also formed as minor products.

Frapper et al. investigated experimentally and theoretically the role of water in a Knoevenagel/hetero-Diels–Alder sequence involving α-methylstyrene **194**, formaldehyde (**195**) and 2,4-pentanedione (**196**) as reactants (Figure 44) [124]. It was experimentally observed that the multicomponent reaction performed in water leads to dehydropyran **197** in higher yield than that carried out in water-miscible organic solvents (e.g., THF, acetonitrile) or water-immiscible solvents (e.g., CHCl_3_ or toluene). On the other hand, the mechanistic studies revealed the role of water in the elimination step leading to **199**, which favors the formation of six-membered transition state **198** and consequently a lower activation free energy barrier. In addition, it was found that owing to the presence of water, the activation free energy barriers of all chemical steps involved in the Knoevenagel/hetero-Diels–Alder sequence were lower than 39 kcal mol^−1^ at 25 °C, thus confirming that the use of catalyst is not necessary for the reaction to occur.

The furan/maleimide Diels–Alder reaction has been used for the covalent functionalization of graphene with polymers; however, in general, long reaction times and high temperatures were required. Recently, Lim and coworkers reported a mild protocol for the covalent direct functionalization of reduced graphene oxide (rGO) by furan/maleimide Diels–Alder click reaction in water [125]. rGO/PSMF hybrids **201** were obtained by reaction of rGO with poly(styrene-alt-maleic anhydride) bearing furfuryl groups (PSMF) **200** under ultrasound irradiation at a frequency of 35 kHz (Figure 45). It is noteworthy that under these conditions, rGO/PSMF hybrids **201** were obtained with 13 wt% grafted PSMF, while for the reaction under conventional stirring conditions over 48 h, the grafting was only 8 wt%, thus demonstrating the ability of sonication to accelerate the Diels–Alder reaction.

In the last decade, the furan/maleimide Diels–Alder click reaction in water has found wide application in the synthesis of biopolymer-based hydrogels. Wei and coworkers reported the synthesis of thermoresponsive hydrogels **204** through a Diels–Alder reaction in water between poly(*N*,*N*-dimethylacrylamide-*co*-furfuryl methacrylate) **202** and *N*-maleolyl alanine poly(ethylene glycol) **203** (Figure 46) [126]. The reaction proceeds efficiently under mild conditions (37 °C) in the absence of catalysts, initiators or coupling reagents, thus providing an alternative to conventional conjugation strategies such as copper-catalyzed azide–alkyne cycloaddition whose intrinsic toxicity limits its application in biological systems. It is noteworthy that the hydrogels **204** are stable in water and that the thermal reversibility was demonstrated since the retro-Diels–Alder reaction could be carried out easily in *N*,*N*-dimethylformamide at higher temperatures (80–100 °C). The aqueous Diels–Alder click reaction was also applied to the synthesis of degradable poly(ethylene glycol)-based hydrogels using maleimide- and furyl-substituted PEG macromonomers as starting materials [127]. In this case, hydrogel degradation occurs within days to weeks at body temperature via retro-Diels–Alder reaction followed by hydrolysis of the maleimide groups. The straightforwardness of the synthesis and the degradability of these hydrogels make them valuable biomaterials for application in controlled drug/protein release or in the area of tissue engineering, where degradation of the biomaterial is often necessary.

In 2011, Shoichet and coworkers disclosed the first synthesis of cross-linked hyaluronic acid (HA) hydrogels based on Diels–Alder click chemistry [128]. The synthetic procedure, involving a clean one-step, catalyst-free, aqueous-based Diels–Alder reaction between furan-modified HA derivatives **205** and dimaleimide poly(ethylene glycol) **206**, led to the formation of cytocompatible HA-PEG hydrogels **207** with potential application in tissue engineering and regenerative medicine (Figure 47).

In the same year, Marra and coworkers reported the synthesis of polysaccharide biodegradable hydrogels **210** for protein encapsulation via the Diels–Alder reaction of maleimide-functionalized HA **208** and furan-functionalized HA **209** using water as solvent medium (Figure 48) [129]. The cycloaddition reaction was selective and efficient for the polysaccharide bioconjugation, allowing the direct encapsulation of positive and negative proteins, lysosome and insulin, respectively, in the biodegradable hydrogels within 40 min of gelation time. Furthermore, it has been shown that proteins can be released from the hydrogel into the local microenvironment in a controlled manner. The furan- and maleimide-functionalized polysaccharide coupling via aqueous Diels–Alder reaction was also explored for the development of controlled drug delivery systems, namely the synthesis of biodegradable hyaluronic acid hydrogels to control the release of dexamethasone [130]. In this case, furan- and maleimide-functionalized hyaluronic acids were used to conjugate the hydrogel, and furan-functionalized dexamethasone was used for the covalent immobilization.

The aqueous furan/maleimide Diels–Alder click reaction strategy was also applied to the synthesis of thermally reversible nanocellulose hydrogels using furan-functionalized nanocellulose fibers and water-soluble oligoether bismaleimides [131]. In this case, the cross-linking reaction was carried out at 65 °C, and the reaction could be reverted at 95 °C via retro-Diels–Alder reaction.

Recently, the synthesis of self-healing pectin/chitosan hybrid hydrogels for drug delivery via Diels–Alder coupling in aqueous medium has also been reported [132]. These polysaccharide biocompatible hydrogels were efficiently obtained through the Diels–Alder reaction of furan-functionalized pectin and maleimide-functionalized chitosan at 65 °C over 5 h, and they showed high swelling properties, pH-responsiveness and cytocompatibility.

Gabilondo’s group reported the synthesis of several biocompatible hydrogels by the Diels–Alder click cross-linking reaction between functionalized furans and maleimides in water, namely methacrylate-based hydrogels [133], methacrylate/polyetheramine [134] and starch/graphene crosslinked hydrogels [135], chitosan-based hydrogels [136] and starch-based nanocomposite hydrogels [137]. The starch-based nanocomposite hydrogels were prepared through the Diels–Alder click reaction between a furan-functionalized starch derivative **211** and a water-soluble PEG-based tetramaleimide **212**, followed by the addition of cellulose nanocrystals as nanoreinforcement (Figure 49) [137]. The reaction, performed at 65 °C over 24 h, afforded hydrogels **213** as solid-like robust materials that keep the shape in their hydrated form. It is noteworthy that the incorporation of cellulose nanocrystals influences the morphology of the hydrogels and the drug delivery performance of the materials.

### 7.3. Catalyzed Diels–Alder Cycloaddition Reactions

The aforementioned MCPRs can be classified into ABC and ACC’, referring to the classical conditions (aniline, aldehyde and nucleophilic alkenes) and the use of aniline and two equivalents of a nucleophilic alkene, respectively. One example of the latter, the synthesis of furano[3,2-*c*]-1,2,3,4-tetrahydroquinolines involving the MCPR between anilines **2** and two equivalents of 2,3-dihydrofuran (**6**), has been recently reported by Fernandes and coworkers (Figure 50) [138]. Using water as solvent and *p*-sulfonic acid calix[4]arene (CX4SO_3_H) as organocatalyst, a series of tetrahydroquinoline derivatives **214** were prepared in moderate to excellent yield, with aniline and 4-halo-anilines providing the target products in the highest yields (85–95%). The catalyst CX4SO_3_H could be reused up to four times while maintaining the catalytic activity. Isotopic labeling experiments involving the synthesis of furano[3,2-*c*]-1,2,3,4-tetrahydroquinoline **216**, supported by NMR and mass spectroscopy data, were crucial for the validation of the reaction mechanism. The reaction seems to evolve through a stepwise sequence via ionic intermediates originating oxonium ion **215**, which undergoes an intramolecular electrophilic aromatic substitution, under CX4SO_3_H catalysis, to afford the final products.

The same group developed a highly efficient and green synthetic protocol for the preparation of julolidines based on the MCPRs approach, using the same catalyst (CX4SO_3_H) and star anise oil (93% content of *trans*-anethol) as dienophile (Figure 51) [139]. A range of substituted anilines **2** reacted smoothly with formaldehyde (**195**) and *trans*-anethol (**76**), in water at 96 °C for 2.5 h, under CX4SO_3_H catalysis giving rise to the diastereoisomeric mixture of julolidines **217** and **218** in yields ranging from 15 to 85%. No significant diastereoisomer excess was observed, albeit the *trans* isomer was the major isomer in all cases. Although both electron-donating and -withdrawing groups were well tolerated, higher yields were attained with anilines bearing electron-withdrawing groups. Furthermore, the easy workup procedure allowed the organocatalyst to be reused up to four times without relevant loss of catalytic activity.

The synthesis of julolidines through the one-pot cascade reaction of aniline derivatives **219** with a mixture of styrene (**220**) and formaldehyde (**195**) using silica sulfuric acid (SSA) as catalyst has also been reported (Figure 52) [140]. Reactions were carried out in water under refluxing conditions for 24 h and led to the desired products **221** as diastereoisomeric mixtures in moderate to good yield. When *o*-substituted anilines were used, tetrahydroquinolines **222** were obtained, although the increase in reaction time (48 h) was required to attain good efficiency.

Beifuss and coworkers developed a straightforward procedure for the diastereoselective synthesis of tetrahydroquinolines based on a domino process involving in situ reduction, imine formation and aza-Diels–Alder reaction (Figure 53a) [141]. Using nitrobenzenes instead of the typical aniline derivatives, the domino process was triggered by in situ reduction of **224** with iron in combination with citric acid as chelating ligand and montmorillonite as catalyst of the Povarov reaction. Thus, the montmorillonite-catalyzed three-component reaction between aldehydes **223**, nitrobenzenes **224** and cyclopentadiene (**39**) in aqueous citric acid at 40 °C, in the presence of iron, gave rise to tetrahydroquinolines **225**/**226** with high *endo*-selectivity and high yields. The catalytic method showed a wide substrate scope, with several substituted nitrobenzenes and aromatic, heteroaromatic and aliphatic aldehydes as suitable substrates. Later, the same group established the intramolecular version of this methodology by using ω-unsaturated aldehydes (e.g., **227**) as substrates (Figure 53b) [142]. Under the same catalytic system, diastereomerically pure *trans*-fused tetrahydrochromano[4,3-*b*]quinolones **228** were obtained from the one-pot domino reduction/imine formation/intramolecular aza-Diels–Alder reaction between nitrobenzenes **224** and ω-unsaturated aldehydes **227**, in yields ranging from 69 to 87%. However, higher temperatures (80 °C) and an increased amount of iron (4 equiv) and montmorillonite (10 wt%) were necessary to attain high levels of efficiency. The observed diastereoselectivity was rationalized as resulting from the *exo*-*E-anti* transition state **229** in the cyclization step, in which both the C=N bond of the diene and the C=C of the dienophile present *E*-configuration (Figure 53b).

More recently, a different methodology for preparing spiro(isoxazolo[5,4-*b*]pyridine-5,5′-pyrimidine) derivatives **232** based on an *L*-proline-promoted one-pot aza-Diels–Alder reaction in water has been disclosed (Figure 54) [143]. This method has the advantage of generating in situ the 5-amino-3-methylisoxazole (**160a**) from readily available 3-aminocrotononitrile (**230**) and hydroxylamine hydrochloride (**231**). Designed experiments were performed and confirmed the formation of isoxazole **160a** and Knoevenagel adduct **166** as reaction intermediates. The presence of *L*-proline seems to be a requirement for attaining high efficiency since carrying out the reaction in its absence led to **232** in decreased yield (29%); however, no information was given regarding the stereochemistry outcome or its role in an eventual asymmetric induction.

Kouznetsov and coworkers have reported a green methodology for the preparation of a series of 4-amido-*N*-yl-2-methyl-tetrahydroquinolines (THQs) based on an acid-promoted domino sequence involving “Mannich-like” reaction/imino-Diels–Alder reaction (Figure 55) [144]. This ABB’ three-component reaction employs one molecule of functionalized aniline **2** and two molecules of cyclic or acyclic *N*-vinyl amides **233** to furnish the target *cis*-2,4-disubstituted THQs **235** in high to excellent yield, in acidified water using sodium dodecyl sulfate (SDS) surfactant as catalyst. The proposed mechanism involves a domino process ran in the proximity of the anionic SDS micelle surface, presenting some charged species and protons (NH and H^+^), initiated by protonation of **233** followed by reaction with amine **2** to give intermediate **234**, which then reacts with the second molecule of **233** to afford the desired products **235**. Consequently, the reaction outcome was strongly influenced by the micellar concentration as well as by the pH reaction medium. In fact, no reaction occurred in the absence of SDS or when the pH was higher (e.g., pH 7). The best results were achieved when using a 12 mM SDS concentration and pH 1. Moreover, this protocol has the advantage of using a micellar aqueous catalyst that is biodegradable and can be reused.

Using a fluorous micellar system in water, several Diels–Alder reactions between typical dienophiles and dienes proceeded in high yields, with remarkable rate acceleration [145]. For example, 9-methylanthracene (**236**) underwent [4 + 2] annulation reaction with *N*-octyl maleimide (**55e**) to give the corresponding cycloadduct **237** in quantitative yield, after only 10 min at room temperature (Figure 56a). The reaction media, composed of 100 mM perfluorohexane (PFH) and lithium perfluorooctanesulfonate (LiFOS)/water (10 mL each), originate a large interfacial area between the fluorous solvent and the water, where the reaction occurs more effectively due to repulsion effects from both media. Furthermore, LiFOS acts not only as a surfactant but also as a supporting electrolyte, enabling the use of electrochemical approaches in the micellar system. Thus, ethyl 3,4-dioxocyclohexa-1,5-dienecarboxylate (**240**), generated electrochemically from ethyl 3,4-dihydroxybenzoate (**238**), was trapped by dienes (e.g., **239**) at the surface of the micelle or at the interface to afford the corresponding Diels–Alder cycloadduct (e.g., **241**) in excellent yield (Figure 56b).

The development of new carbocatalysts which are eco-friendly and sustainable has attracted great attention. Owing to their easy preparation from inexpensive graphite, graphene and graphene oxide (GO) have emerged as efficient carbocatalysts for several transformations in organic synthesis, namely the Diels–Alder reaction. De and coworkers have disclosed the GO-promoted Diels–Alder reaction between 9-hydroxymethylanthracene (**54**) and *N*-substituted maleimides **55** in an aqueous medium at room temperature leading to the corresponding cycloadducts **242** in moderate to excellent yields (Figure 57) [146]. The catalytic protocol has a wide substrate scope, and the GO catalyst can be easily recovered by centrifugation or filtration and reutilized up to four times without significant loss of activity. A three-step mechanism was proposed involving the initial binding of the hydroxyl groups and the aromatic rings to the GO surface by weak hydrogen bonding and π–π interactions. Subsequently to the product formation, the anthracene moiety loses its planar structure as well as the partial aromaticity leading to the weakness of the π–π interactions and displacement of the product by another molecule of reactant. Experimental calculations were carried out and corroborated the proposed mechanism.

### 7.4. Asymmetric Diels–Alder Cycloaddition Reactions

Zhang and coworkers reported a rate-accelerating effect in the asymmetric organocatalytic Diels–Alder reactions between α,β-unsaturated aldehydes **40** and cyclopentadiene (**39**), catalyzed by C_2_-symmetric bipyrrolidine **243** and HClO_4_ in aqueous reaction medium (Figure 58) [147]. In contrast with the reactions performed in organic solvents, the reactions carried out in water were completed within only 2–3.5 h, furnishing cycloadducts **244** in high yields with good enantioselectivity and moderate *exo*-selectivity. Moreover, the catalyst **243**. 2HClO_4_ could be recovered by diethyl ether extraction and reused directly up to four times while retaining its catalytic activity. DFT calculations and X-ray studies corroborated the proposed mechanism which involves the formation of a C_2_-symmetric diiminium intermediate (e.g., (*E*,*E*)-**245**), with two conjugated iminiums facing each other, corresponding to two *Si* faces giving rise to the same enantiofacial discrimination in the cycloaddition step.

A poly(methylhydrosiloxane) (PMHS)-supported chiral organic catalyst derived from MacMillan’s imidazolidin-4-ones (e.g., **246**) also has the ability to promote successfully Diels–Alder cycloaddition reactions in aqueous medium with a high degree of enantioselectivity (up to 93% ee), albeit with low *exo/endo* selectivity (Figure 59) [148]. The cycloaddition reaction between *trans*-cinnamaldehyde (**40a**) and cyclopentadiene (**39**) was investigated under different catalytic reaction conditions, with higher yields and enantioselectivities being achieved when a preformed PMHS-supported HBF_4_ salt of **246** was used in water. Nevertheless, it was observed that when using only water as solvent, the catalyst activity decreased dramatically after the second recycle run, whereas a 95:5 acetonitrile:H_2_O mixture proved to be the ideal solvent system to ensure the recycling of the PMHS-supported catalyst.

In the same year, Wang and coworkers disclosed a straightforward route to new heterogeneous organocatalysts based on the functionalization of hollow-structured phenylene-bridged periodic mesoporous organosilica (PMO) spheres with MacMillan catalyst and investigated its use in the aforementioned Diels–Alder reaction [149]. The cycloaddition reaction performed in water using 20 mol% of catalyst, in the presence of TFA, led to cycloadducts **244a** in excellent overall yield (98%) and high enantiomeric excess (81% for *endo*-adduct and 81% for *exo*-adduct). Furthermore, the H-*Ph*-PMO-Mac catalyst could be reutilized for up to seven catalytic cycles without a significant loss of catalytic activity.

The “on-water” organocatalyzed [4 + 2] cycloaddition reaction of acyclic enones with nitro dienes and allylidene malononitriles gave access to a plethora of functionalized chiral cyclohexanones, using cinchona alkaloid-based primary amines (e.g., **249** and **252**) and benzoic acid as the organocatalytic system (Figure 60) [150,151]. Thus, unsaturated methyl ketones **247** reacted smoothly with primary amine catalyst **249** in the presence of benzoic acid to give the corresponding enamine, which undergoes an *endo* [4 + 2] cycloaddition with nitro dienes **248** to yield the corresponding 3,4,5-trisubstituted cyclohexanones **250** in good yield, with good diastereoselectivities and excellent enantioselectivities. The reaction outcome was not significantly affected by the nature and positioning of the aryl substituents of both reactants. Interestingly, mechanistic studies revealed that the kinetic *endo*-cycloadduct initially formed with an α-nitro-bearing center, epimerized under the reaction conditions to thermodynamically more stable diastereoisomer **250** with a β-nitro-bearing center. On the other hand, the use of allylidene malononitriles **251** as dienophiles led to the enantioselective synthesis of cyclohexanones **253** with two stereocenters and an all-carbon quaternary center. The highest enantioselectivity was achieved from the reaction of benzylidene acetone with allylidene malononitrile bearing a nitro-substituted aryl group (82% ee). Moreover, allylidene cyanoacetates were also suitable dienophiles for these transformations.

DNA-based hybrid catalysts combine the catalytic power of a metal complex with the unique helical chirality and chemical stability of DNA and have been employed successfully in various asymmetric transformations, namely the asymmetric Diels–Alder reaction. A solid-supported DNA (st-DNA/S1), prepared from purified salmon testes DNA (st-DNA) and ammonium-functionalized silica (S1), has been used in the copper(ii)-catalyzed Diels–Alder reaction of 2-azachalcone **254a** and cyclopentadiene (**39**) in water (Figure 61) [152]. The catalytic protocol involves the mixing of a Cu(dmbpy) complex with st-DNA/S1 in a MOPS buffer solution, the subsequent addition of the reactants and the continuous rotation of the reaction mixture at 5 °C for 3 days. The target cycloadduct **255** was obtained with excellent *endo*-selectivity (*endo*/*exo* ratio of 99:1) with 99% conversion and 94% ee. Performing the reaction with unsupported st-DNA led to **255** with a slight increase in the ee (99%) and similar conversion (93%), whereas the controlled experiment using only S1 without st-DNA afforded **255** in low conversion (14%) and neglectable ee (<3%). In addition, the catalyst st-DNA/S1 was recycled and reused for 10 cycles affording conversions above 93% in each cycle and ee’s ranging from 88 to 94%, thus demonstrating its utility as a reusable chiral source.

In another example of bioinspired catalysts, Liskamp and coworkers disclosed the asymmetric copper(ii)-promoted Diels–Alder reaction in water using tris-histidine-containing triazacyclophane (TAC), a scaffold that mimics the structure of the tris-histidine triad metal-binding site found in several metalloenzymes (Figure 62) [153]. Due to their ability to coordinate efficiently with Cu^II^ in water, 2-azachalcone **254a** and α,β-unsaturated 2-acylimidazole **254b** were selected as substrates for the Cu(ii)-catalyzed cycloaddition with cyclopentadiene (**39**). Among the several TAC-based ligands screened, compounds **256** and **257** bearing three *D*- and *L*-histidine residues, respectively, directly attached to the TAC scaffold and acetylated α-amino groups, afforded selectively the target products **255** with the highest ee (up to 55%)**.** Using ligand **256**, cycloadduct **255** with *SS* configuration was obtained, whereas ligand **257** induced the selective synthesis of its enantiomer. Moreover, attempts to further improve the enantioselectivity through the insertion of additional amino acid residues on the *N*-termini of the histidine moieties, as well as between the TAC scaffold and the histidine residues, resulted in significantly decreased ee.

### 7.5. Total Synthesis

The generation of high structural complexity from simple starting materials through the creation of two C–C bonds, at least two rings and up to four stereoisomers in a single step makes the intramolecular Diels–Alder a powerful synthetic tool in the synthesis of complex natural products such as verrubenzospirolactone, a meroterpenoid isolated from the soft coral *Simularia verruca* [154,155,156]. The total synthesis of verrubenzospirolactone **262**, featuring a pentacyclic structure comprising a spirocyclic butenolide and five contiguous stereocenters, was successfully accomplished in five steps starting from readily available methylhydroquinone (**258**) and citral (**259**) (Figure 63a) [157]. The key step of this synthesis involved an unusual intramolecular Diels–Alder reaction between the 2*H*-chromene and diene moieties of conjugated triene (*Z*)-**261** which upon heating “on-water” at 50 °C afforded verrubenzospirolactone **262** in 61% yield, along with its C9 epimer **263** in 10% yield. The former is the intramolecular Diels–Alder cycloadduct obtained from the *s*-*cis* conformation of (*Z*)-**261** via an *exo* transition state, whereas the latter resulted from the *exo* intramolecular Diels–Alder reaction of the *s*-*cis* conformation of (*E*)-**261**. Starting from a more functionalized aldehyde bearing a triene unit, a quadruple cascade reaction consisting of a Knoevenagel condensation/oxa-6π-electrocyclization/Diels–Alder/oxa-Michael sequence was developed, leading to the formation of a 3:1 mixture of hexacycles **265** (epimers at C11) in 46% yield (Figure 63b).

The total synthesis of natural product delitschiapyrone A, featuring an impressive 6/6/5/7/6 pentacyclic ring system with five contiguous stereocenters, has also been recently disclosed [158]. This natural product was isolated from the fungus *Delitschia* sp. FL1581 and exhibits cytotoxic activity against several cancer cell lines (IC_50_: 12.3–35.5 µM). This challenging synthesis involves a seven-step sequence in which the Diels–Alder reaction of juglone derivative **266** with diene **267** plays a crucial role (Figure 64). The Diels–Alder reaction, followed by concomitant α-ketol rearrangement and cyclic hemiacetalization, proceeded smoothly at 35 °C in water, leading to the desired cycloadduct **270** in 75% yield, with exclusive regioselectivity and stereoselectivity. To rationalize the observed regio- and diastereoselectivity, the energies and coefficients of the frontier orbitals of **266** and **267** were calculated and the DFT-optimized structure of the preferred *endo* transition state of the Diels–Alder reaction was determined. The presence of the hydroxyl of diene **267** seems to be a requirement to attain selectivity in the Diels–Alder reaction since the reaction of **266** with *O*-acetyl-**267** led to a complex mixture.

## 8. Conclusions

The Diels–Alder reaction remains one of the most commonly employed reactions for the rapid construction of carbocyclic and heterocyclic compounds, including the synthesis of natural products. In this review, the most significant advances on the Diels–Alder reaction in environmentally benign solvent systems were highlighted.

The fast-growing green chemistry research area has driven successful developments on this topic, upgrading this traditional reaction to the present sustainable needs. Notably, high levels of chemical efficiency and selectivity were achieved in most of the described methodologies, with the use of these sustainable solvent media leading to shorter reaction times and simpler workup procedures than the use of conventional organic solvents. Additionally, the Lewis acid catalysis and organocatalysis combined with nonconventional heating methods have broadened the scope of these methodologies. Despite the recent progress, further developments on sustainable enantioselective synthetic approaches are required. The development of highly active catalysts, namely functionalized heterogeneous catalysts with well-defined structures, suitable to a wider range of substrates and solvents, is needed to overcome some of the current challenges. Therefore, innovative strategies for the design of new catalytic systems with enhanced properties can be expected in the near future.

## Data Availability

Not applicable.

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
