# Peer review of "Diels–Alder Cycloaddition Reactions in Sustainable Media"

_molecules, 2022, doi:10.3390/molecules27041304_

Round 1

Reviewer 1 Report

The manuscript submitted by Soares and Pinho e Melo mainly reviewed the relevant advances on sustainable Diels−Alder reactions since 2010. Various environmentally benign solvent systems were discussed, that is, bio-based derived solvents such as glycerol and gluconic acid, polyethylene glycol, deep eutectic solvents, supercritical carbon dioxide, water and water-based aqueous systems. Subjects such as method’s scope, efficiency, selectivity and reaction mechanism, as well as sustainability, advantages and limitations of these reaction media were addressed. In light of the information that the Diels-Alder reaction remains one of the most commonly employed reactions for the rapid construction of carbocyclic and heterocyclic compounds, along with the fact that the recent green chemistry research has upgraded Diels-Alder reactions to fulfill the present sustainable needs, I believe the readers of this journal might show interests in reading this manuscript if it is published. Thus, the publication of this review article on the journal of Molecule is recommended. Some minor suggestions for the authors are listed below:

  1. In Scheme 30 on page 22, a double bond is missing for the structure of compound 133.
  2. In Scheme 34 on page 24, a nitrogen atom is missing for the structure of compound 151a. Also, the cyclopentadienyl moiety shown in compound 155 should be the phenyl group.
  3. Line 906 on page 32, “The latter were prepared though the---”, though changes to through.

Author Response

We thank the reviewer for his/her positive appreciation of our manuscript. As indicated by the referee, minor inaccuracies and typographical errors detected have been corrected (highlighted in yellow):

  1. In Scheme 30 on page 22, a double bond is missing for the structure of compound 133. DONE
  2. In Scheme 34 on page 24, a nitrogen atom is missing for the structure of compound 151a. Also, the cyclopentadienyl moiety shown in compound 155 should be the phenyl group. DONE
  3. Line 906 on page 32, “The latter were prepared though the---”, though changes to through. DONE

Reviewer 2 Report

The review paper is addresses recent advances (since 2010) in the Diels−Alder reaction in environmentally benign and sustainable solvent systems. The paper presents [4+2] cycloaddition reactions performed in biomass derived solvents, polyethylene glycol, organic carbonates, deep eutectic solvents, supercritical CO2 and water. The paper is well organized and written and will be interesting to a broad audience of synthetic chemists.

Some minor inaccuracies and typographical errors were detected.  

I suggest that the paper can be accepted for publication

p.3, line 93 , Schemes 3 and 4, and throughout the manuscript: should be “oxa”, not “oxo”

Scheme 11 should be “Frutose” not Fructose

Scheme 16 and throughout the manuscript: should be “azaalkene”, not  “azoalkane”

I suggest that the paper can be accepted for publication

Author Response

We thank the reviewer for his/her positive appreciation of our manuscript. As indicated by the referee, some minor inaccuracies and typographical errors detected have been corrected:

1. p.3, line 93 , Schemes 3 and 4, and throughout the manuscript: should be “oxa”, not “oxo”. DONE

2. Scheme 11 should be “Frutose” not Fructose. "Frutose" was replaced by frutose. DONE

3. Scheme 16 and throughout the manuscript: should be “azaalkene”, not  “azoalkane”. Reply: We respect the point of view of the reviewer, but we do not agree with it. “Azaalkene” is a compound with only one nitrogen atom replacing the carbon in the double bond, whereas “azoalkene” is a 1,2-diaza-1,3-diene, thus having two nitrogen atoms, which is the case of compounds in schemes 16 and 32. Thus, we could only replace “azo” by “diaza”, but we kept it as is was. Several examples of the use of this terminology can be found in recent literature (e.g. J. Org. Chem. 2020, 85, 18, 11812; Chem. Commun. 2019, 55, 6672; Angew. Chem. Int. Ed. 2020, 59, 648). No changes made to the manuscript.

Reviewer 3 Report

The manuscript is a comprehensive review of Diels-Alder (DA) cycloaddition reactions in environmentally friendly, bio-based solvents, which have been published in the literature in the last ten years. There are 64 schemes and 158 references in the manuscript, which is neither too little nor too many, but more importantly, the examples are representative and well selected from appropriate sources, relevant journals in the field. The schemes are very clear and provide all the most important information in the caption, and the scheme itself, about the shown DA reaction (including the number of examples). Particularly commendable is the use of coloured bonds to highlight parts of the structures involved in the reaction, red colour for marking stereochemistry, somewhere curly arrows for mechanisms, and the special emphasis on the used green solvents (marked green or blue for “on-water” DA reactions) in the schemes. Whenever an organic solvent has been used in a part of the reaction, this is especially emphasized in the text and / or in the scheme, which I also think is important.

As expected and most rationally, after the Introduction, the manuscript is divided into sections according to the solvent used in the covered DA reactions: 2. Bio-based solvents (2.1. Glycerol, 2.2. Gluconic acid), 3. 3. Polyethylene Glycol (PEG), 4. Organic Carbonates, 5. Deep Eutectic Solvents, 6. Supercritical Carbon Dioxide and 7. Water. The “Water” section is justifiably the longest and first it gives an insight into the water effect through experimental and theoretical studies, then divides into catalyzed and non-catalyzed DA reactions, followed by an overview of asymmetric DA reactions and finally in the end describes a couple of examples of total syntheses that involve “on-water” DA reactions as the key steps. All in all, the examples in the review are well selected and show well, in addition to various green solvents, a sufficient number of different reactants and products, stereochemical outcomes, reaction conditions and energy sources. The conclusion is brief, but critically highlights what seems most important at the moment to overcome the main challenges.

The manuscript is nicely written, easy to read and the review is done very studiously, so I believe it will be helpful to experts in the field, but also interesting to other readers who deal with organic syntheses in general.

Author Response

We thank the reviewer for his/her positive appreciation of our manuscript.